# CNN-LSTM based emotion recognition using Chebyshev moment and K-fold validation with multi-library SVM

**Samanthisvaran Jayaraman[1], Anand Mahendran[2]***

**1** School of Computer Science and Engineering, Vellore Institute of Technology, Vellore 632014, Tamil Nadu, India, **2** School of Computer Science and Engineering, Vellore Institute of Technology, Chennai 600127, Tamil Nadu, India

\* manand@vit.ac.in

## Abstract

Human emotions are not necessarily tends to produce right facial expressions as there is no well defined connection between them. Although, human emotions are sponta-neous, their facial expressions depend a lot on their mental and psychological capacity to either hide it or show it explicitly. Over a decade, Machine Learning and Neural Networks methodologies are most widely used by the researchers to tackle these challenges, and to deliver an improved performance with accuracy. This paper focuses on analyzing the driver's facial expressions to determine their mood or emotional state while driving to ensure their safety. we propose a hybrid CNN-LSTM model in which RESNET152 CNN is used along with Multi-Library Support Vector Machine for classification purposes. For the betterment of feature extraction, this study has considered Chebyshev moment which plays an important role as it has a repetition process to gain primary features and K-fold validation helps to evaluate the models performance in terms of both training, valida-tion loss, training, and validation accuracy. This study performance was evaluated and compared with existing hybrid approaches like CNN-SVM and ANN-LSTM where the proposed model delivered better results than other models considered.

## Introduction

Emotions are highly natural for any human based on the situation they are in. Thus, they has have a significant effect on humans behavior that is likely leads to many different con-sequences, both positively and negatively. Especially, in case of drivers, it is very important and crucial to identify and monitor the emotional state of drivers, not only to asses his/her health and psychological conditions, but also to avoid both auto road and air crash accidents. Healthy mind always lead to healthy life [1]. It is very much evident from the history of inci-dents that has led to fatal accidents which took hundreds and thousands of lives due to the mental illness or emotional conditions of the drivers, captain or pilots, especially, on the roads and in the air. Most of these accidents happen due to the driver's health and emotional level [2].

**Data availability statement:** The data for this study can be found at https://www.kaggle.com/datasets/msambare/fer2013.

**Funding:** The author(s) received no specific funding for this work.

**Competing interests:** The authors have declared that no competing interests exist.

Though automotive companies aims to launch autonomous vehicles over a decade, fatal accidents at regular intervals has decreased the overall public interest in autonomous cars as both safety and reality matters to the end customers has fatal accidents at Pheonix, in 2018 and, Tesla in 2022 and 2023 has decreased the overall public interest in autonomous cars as both safety and reality matters to the end customers [3].

Currently, there are more than 25 companies offering autonomous vehicles yet more than 130 accidents reported in 2022. Yet, these automotive companies are still confident of achieving accident free autonomous cars by the end of 2025 [4]. But, the question will remain is determining where liability rests in case of accidents.

Figure 1 depicts the expected percentage of cars with autonomous driving worldwide between 2015 and 2025 reported by statista report 2023 and this report considered three levels of driving, namely, no autonomous driving, partly assisted driving, assisted driving and highly autonomous driving [5].

According to report by Virginia Tech, the following are the driver's actions that likely lead to accidents. Table 1 represents the report findings.

Mercedes and MAN (Metropolitan Area Network) offer driver's concentration control system such as falling in sleep while driving and although car producers ensures safety with their inbuilt systems, no statistical data available to indicate how these systems affect the reduction of accidents [6,7].

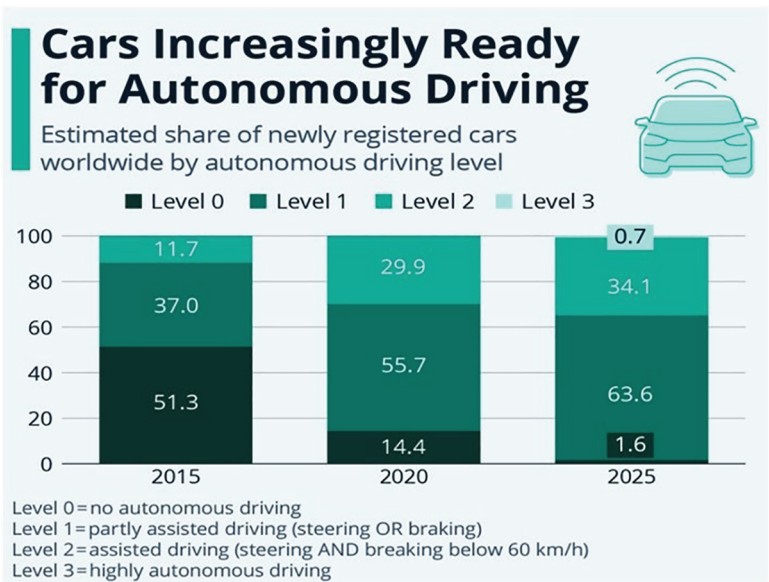

**Fig 1. Percentage of cars with Autonomous Driving by 2025. (Source: www.statista.com.)**

**Table 1. Driver Actions Lead to accidents [6]**

| Cause of an accident | Percentage |
|---|---|
| Under liquor/drugs | 35 |
| Using mobile devices | 20 |
| Reading/writing | 16 |
| Anger/sadness | 14 |
| Chasing an object | 09 |
| Other reasons | 06 |

Facial Expression Recognition (FER) has increasingly become a more challenging and multifaceted research topic among the researchers in the last decade. FER plays crucial role in various applications such as safe driving, health care, crime identification and so forth, further the credibility of such methods has led to intelligent outcomes in the field of Human Computer Interaction (HCI). Deep Learning (DL) and Artificial Intelligence (AI) based FER methods helped to process the edge modules and real time process with improved efficiency [8].

Figure 2 represents the components of x-Emotion subsystem discussed in [9]. Emotions might not be based on a single factor as it is: context to impressions, context to objects, state of an agent and mood. The impressions are generally short in time with perceived features; objects are also short in time but deals with expressive sub-emotions. The state represents the classical emotion and finally mood represents the emotion that lasts for long time [9,10] Illumination,pose, background and camera view point of a source image have significant effect on FER performance in terms of accuracy that might lead to occlusion or misalignment. In addition, facial expression is nothing but a response (stimulus and response) that imitates an emotion.

Figure 3 represents the facial landmarks of a person with numbers. Facial landmarks are vital that consists of visually salient points which basically include end of eyebrows, end of mouth and end of nose as landmark points. Either local texture of these landmark points or pairwise positions of any two landmark points are used as features.

Figure 3 has 64 landmark points and for each number the description is given in Table 2 as primary and secondary landmarks [10].Convolution Neural Networks (CNN) have shown great potential in recognizing facial expression due to their computational efficiency and feature extraction capability, and CNN is the most commonly used deep model for Facial Expression Recognition (FER). The main objective of these methods is to improve classification accuracy [11,12].

The primary goal of this work is to improve the accuracy of classification in FER using deep learning techniques by detecting different sort of emotions in FER2013 dataset. We have chosen this dataset as it encompasses the difficult naturalistic conditions and challenges. Importantly, human emotions in FER2013 dataset are estimated to be 65.5% [13].

This research work is organized as follows in this paper. Introduction, provides basic need and understanding of Facial Expression Recognition (FER), Related Work, discusses about

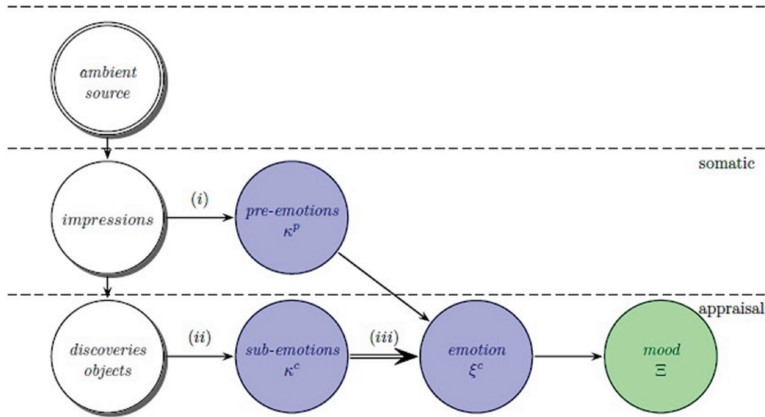

**Fig 2. Emotional components and basic relationships [9].**

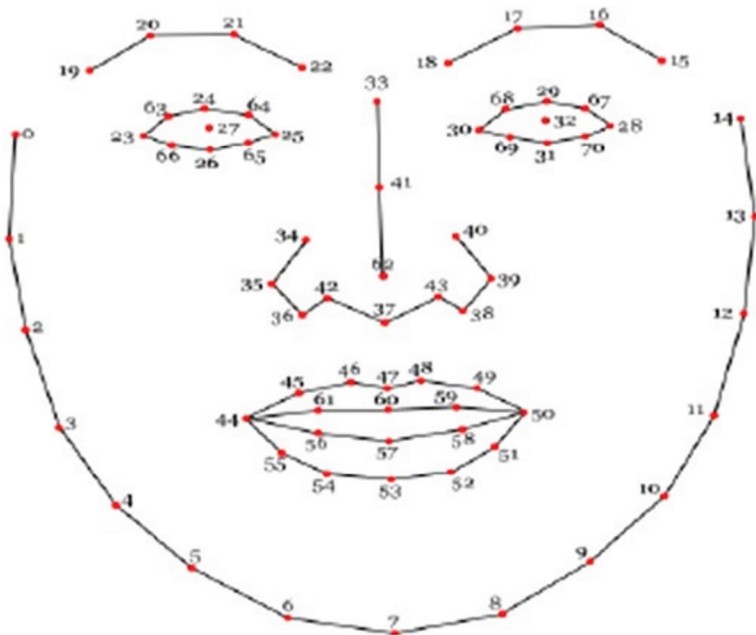

**Fig 3. Facial Landmarks to be extracted from face [11].**

**Table 2. 64 primary and secondary land marks and its definitions [10]**

| Primary Landmarks | | Secondary Landmarks | |
|---|---|---|---|
| Number | Definition | Number | Definition |
| 16 | Left Eyebrow Outer Corner | 1 | Left Temple |
| 19 | Left Eyebrow Inner Corner | 8 | Chin Tip |
| 22 | Right Eyebrow Inner Corner | 2–7, 9–14 | Cheek Contours |
| 25 | Right Eyebrow Outer Corner | 15 | Right Temple |
| 28 | Left Eye Outer Corner | 16–19 | Left Eye Brow Contours |
| 30 | Left Eye Inner Corner | 22–25 | Right Eye Brow Contours |
| 32 | Right Eye Inner Corner | 29,33 | Upper Eyelid Centers |
| 34 | Right Eye Outer Corner | 31,35 | Lower Eyelid Centers |
| 41 | Nose Tip | 36,37 | Nose Saddles |
| 46 | Left Mouth Corner | 40,42 | Nose Peaks (Nostrils) |
| 52 | Right Mouth Corner | 38–40,42–45 | Nose Contours |
| 63–64 | Eye Centers | 47-51,53–62 | Mouth Contours |

various related works relevant to this research study. Proposed approach, introduces a hybrid approach by integrating CNN-LSTM-RESnet152 along with Chebyshev moment and some other methods. Experimentation results, presents the experimentation results obtained and the same is compared with some other works that are relevant to this study in detail. Finally, concludes the study findings and our future research work.

## Related work

In this Related work we discuss about Facial expression of any human tends to convey their mood or mindset. Facial Expression Recognition has increasingly become a hot research topic among the researchers over the decade, although it has become an active research topic,

the key challenge comes from ambiguity and errors when mapping the emotional states to the factors that can be used to detect them [14] Convolution Neural Networks (CNN) is introduced in FER [15,16] which used raw images as an input rather than hand-coded features. CNN is capable of learning a set of pre-defined features with its alternated types of layers such as convolution layers, sub-sampling layers to come up with best classification. The inherent structure of CNN makes it best suitable for image based research. In [17], a different CNN structure was proposed which processed most critical features of an image such as nose, eyebrow and mouth and forwarded it to the next layers to minimize the redundancy of features that are already learnt. The same is followed between the filters of the same layer. In [18], self collected facial images were used with a standard CNN with 2 convolution-pooling layers and in [19], a stacked convolutional auto-encoder with weights was proposed and trained the model raw images. This approach has out performed the results obtained of CNN with random initialization. In [20], a graph based CNN was introduced to detect the landmark features of facial images and the results of this approach was proved to be effective.

In [21], the authors have proposed hybrid CNN-LSTM approach where CNN extracts spatial characteristics of single frame and Long Short Term Memory (LSTM) to extract temporal data of multiple frames. This study was aimed to reduce the margin-based loss instead of cross entropy loss. In [22], to better the categorization of images the authors have studied multi-level features where they intensely combined the hierarchy of characteristics of images, and the model was tested on FER2013 dataset but the performance were similar to the existing approaches.

In [23], the authors have tried to overcome the issue of over-fitting by effectively utilizing dropout, regularization and data augmentation in CNN model and, to prevent gradient vanishing and exploding the authors have employed batch normalization. The study has used FER2013 dataset and produced better results. In [24], the proposed approach have adopted Stochastic Gradient Descent (SGD) along with CNN for optimization purposes by updating the parameters of CNN model on the basis of gradient of a single data point. In [25], the authors have used three separate CNN and also trained them separately. Finally, they ensemble them to improve the performance in terms of accuracy, and also has achieved 62.44% with their proposed model. Attention mechanism based CNN was proposed in [26] with end-to-end deep learning framework, this model has achieved 70.02% accuracy.

In [27], the study has used Support Vector Machine by replacing softmax layer for classification purposes that employed deep neural network. The results were much improved with the classification accuracy of 71.20 %. In [28], performances of three different architectures were compared, namely, VCG, Inception and RESNET where the results have showed that VCG with 72.7%, RESNET with 72.4% and Inception with 71.6% of accuracy. In [29], the authors have proposed face2face nodes based on novel graph model which does not require facial landmarks and this model had two components patch embedding with multi-scale feature fusion and relation-aware dynamic graph CNN. This model was mainly aimed to overcome the drawback of CNN's struggle to capture structural correlations and redundant correlation issue of Vision Transformers (ViT's). by incorporation K-nearest neighbors algorithm and relation-aware graph convolution operator, the model has achieved the following accuracies of 91.41%, 91.02%, and 66.69% on FERPlus, RAF-DB and AffectNet datasets, respectively.

In [30], a novel cross-hierarchy contrast (CHC) which employs contrastive learning and this model mainly regularizes feature learning and to improve global representation of facial images expression. A more robust feature representation was obtained using CHC

and backbone network by integrating global features through fusion. The experiments were conducted on six widely used datasets, namely, CK+, FER2013, FER+, RAF-DB, AffectNet and JAFFE. In [31], for accurate emotion detection from video clips, a novel cross attention and hybrid feature weighting network was proposed. This approach had three main components: Dual Branch Encoding (DBE), Hierarchical- attention Encoding (HAE) and Deep Fusion block (DF). In addition, recalibration block and adaptive attention block are integrated with HAE block which revises the feature map of each channel and to optimize fusion feature weights using hybrid weighting operation. Finally, DF block integrates all the features to predict the individual emotional state. The model was experimented on CAER-S dataset and results have shown that the model performed well on the video clips considered for this study.

In [32], the authors have proposed a CNN model with two approaches, namely, Transfer learning which is pre-trained with Inception-V3 and MobileNet-V2 and Taguchi method to improve the hyper parameter setting robustness. The experimentation was conducted of JAFFE and KDEF datasets with ten expression classes, where the proposed model has achieved an accuracy of 96% and F1-Score of 0.95%. In [33], DCNN model was proposed with pre-trained models like EfficientNet, ResNet and VGGNet along with Haar face classifier and the experiment was conducted on video streams unlike static images. The proposed model resulted in 82% accuracy on FER2013 dataset. In [34], Anti-Aliased Deep Convolution Network (AA-DCNN) was proposed to detect eight emotions from static image datasets like CK+, FER2013, JAFFE, and RAF. The main objective was to analyze how anti-aliasing can enhance expression recognition with down-sampling layers. This study has achieved highest accuracy on CK++ with 99.26%, and lowest with RAF at 82%.

In order to improve the accuracy, we aim to combine CNN-RESNET152-LSTM along with Chebyshev, K-fold Validation, and Multi-Library SVM is used for classification purposes, and we also reduce the FER2013 dataset from 35000 images to 7074 images on the basis of image brightness, resolution, contrast and clarity in pose to improve the performance of our proposed model. Moreover, this study has only considered the images, not videos. Yet, there have been some works which have considered video clips. Those works has been referred a it can guide and direct the researcher towards future.

## Motivation of this work

The motivation of this research study was to avoid unnecessary incidents, i.e., could possibly be an accident due to driver's casualness. Though, so many existing models or approaches proposed thus far, this study was aimed to have a progress on this research field, considering the combinations of neural networks, deep learning methods and models. Driver's consciousness is very much important in ensuring driver's safety, also the pedestrians or fellow travelers. Thus, this study aims to achieve the same with following proposed combinations.

## Proposed approach

In this proposed approach, we propose a hybrid CNN-LSTM based network model for a better facial expression recognition with Chebyshev Moment and k-fold validation to verify the accuracy during training and testing phase along with RESNET 152 CNN to tackle sensitivity and over fitting issue of basic CNN model. various components, methods and algorithms.

## Dataset preparation

**FER 2013 dataset:** This dataset is originally introduced during ICML2013 (Challenges in Representation Learning) in 2013. Google image search API has collected a large scale and unconstrained database automatically and later on wrongfully labeled frames were removed and cropped regions were adjusted. It is a known fact that CNN takes input of the same size, since the dataset has images with different size and resolution; the images in FER2013 are resized to 48*48 pixels for experimentation convenience. The shrinking of images means deformation of features and patterns inside the images, thus the size is kept as 48*48 pixels. This dataset represents seven common expressions of a human (anger, disgust, fear, happiness, sadness, surprise and neutral) with 28709 training images, 3589 validation images, and 3589 test images [18]. The raw FER2013 data must be prepared before being given as input to the CNN and the images need to be converted into an array of numbers for the CNN model. The following are the basic issues while using any dataset.

**Imbalance:** The dataset must have a group of class images unbiased. For example, a dataset has 3000 images for happy emotion and 300 for sad, in such case, the designed model results will be biased towards happy emotion. To tackle this issue, data augmentation can be considered which performs crop, flip and padding [11].

**Contrast variation:** The contrast of the images in the dataset may vary as they could have taken at different seasons, Scenes and background. So, the face of the image needs to be focused is very important.

**Intra-class variation:** To avoid this issue, all images that are not human faces to be removed in order to improve the model performance, thus accuracy can be maintained.

**Occlusion:** This problem occurs when part of the face is covered while the person is/was pictured, i.e., wearing sunglasses, sunlight covering one side of the face or left eye or right eye. Table 1 indicates that eye and nose have primary features in extracting and recognizing a person's emotion. Therefore, occluded images need to be removed from the dataset to recognize emotions.

Thus, manual filtering was done on the FER2013 dataset to overcome the above mentioned challenges, as a result, a total of 7074 images were selected based on contrast, brightness, angle of image. The selected images corresponds to five classes, namely, angry=966 images, fear=859 images, happy=2477 images, neutral=1466 images and sad=1326 images. Thus, out of 35000 images of FER2013 dataset, only 7074 were selected for this study based on study [11].

## Framework model of the methodology

Figure 4 represents the overview of the proposed methodology. As mentioned in the above Dataset Preparation, manual filtering is performed to minimize the FER2013 dataset images from 35000 to 7074 in order to test the images with high contrast, brightness, angle of image. Manual filtering is undertaken to avoid blur and very low quality images present in FER2013 dataset. This study has inherited the data set used in [11]. Feature extraction is performed using Gaussian Filter to smooth out facial imperfections on images and automatic censoring was not performed with this filter as it reduces the image quality which is followed by Chebyshev Moment to extract rich and primary features from the images based on the landmarks provided in Table 2. Chebyshev has a repetition process to gain primary features from the input dataset images as this method is very much effective since it computes mean and standard deviation to determine where most of required image features or data fall within the image i.e. primary and secondary features [35].

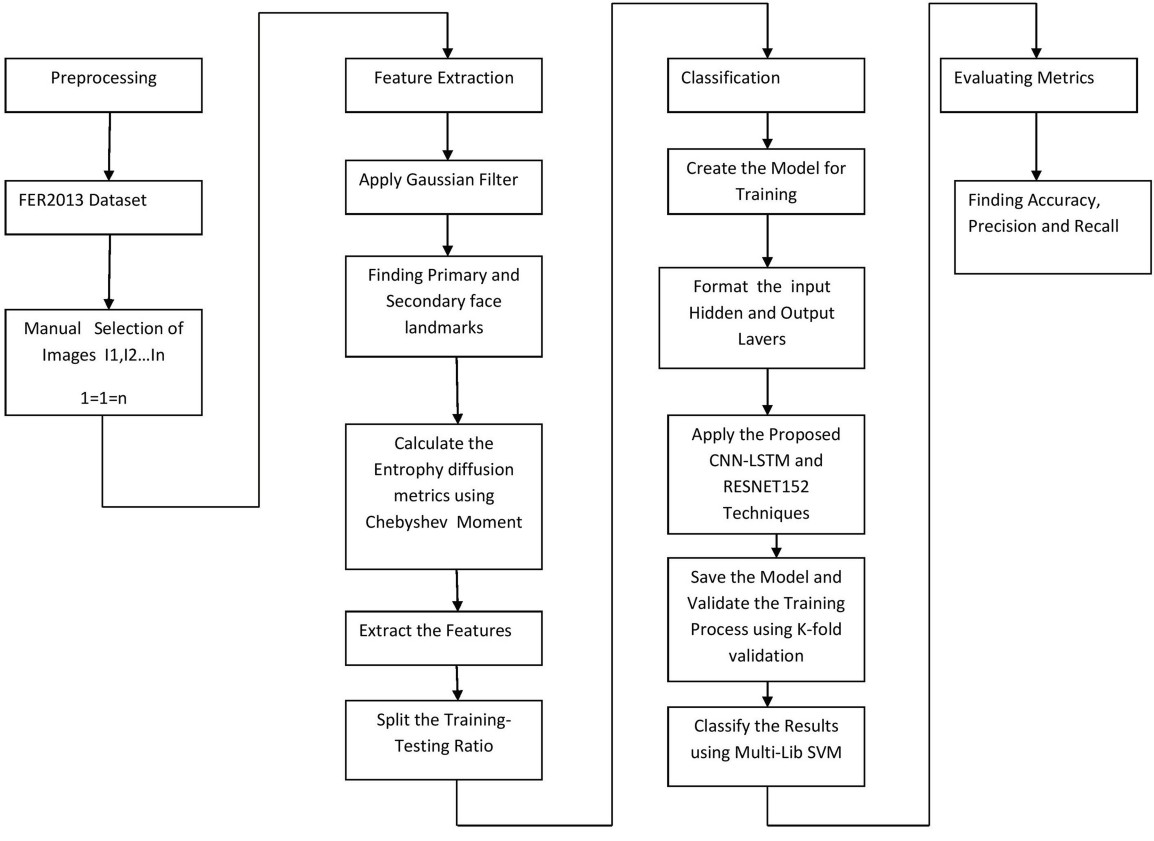

**Fig 4. Overview of proposed work architecture.**

Gaussian filter is very effective in edge detection and also keep the edges relatively sharp which is also faster than median filter and also in removing the noise from the images. In case of box filter, the sharp cutoff leads to noisier output image with less well localized frequency response. Since, Gaussian filter create scale-space representation of an image with qualitatively reduced noise, object recognition can be enhanced [36]. Training and testing ratio is determined and the same is fed as input to the training model of CNN-LSTM-RESNET152 and the model is validated using K-fold validation. The output is classified using Multi-library SVM classifier.

## Feature extraction: Chebyshev moment

For the betterment of feature extraction, this study has considered Chebyshev moment which plays an important role as it has a repetition process to gain primary features from the input dataset images as this method is very much effective since it computes mean and standard deviation to determine where most of required image features or data fall within the image, i.e., primary and secondary features.

Let's consider a moment with the order of $r$ and repetition of $s$ for an undertaken image size $N \star N$ with $m = (N/2 + 1)$ can be written as

$$K_{rs} = \frac{1}{2\pi\rho(\rho,m)} \sum_{p=0}^{m-1} \sum_{\theta=0}^{2\pi} t_\rho(p) * e^{-js\theta} * f(p,\theta) \qquad (1)$$

In the above equation, $t_\rho(p)$ denotes the orthogonal basis Chebyshev polynomial function of an image size of $N \star N$.

$$t_0(x) = 1 \tag{2}$$

$$t_1(x) = \frac{2x - N + 1}{N} \tag{3}$$

$$t_r(x) = \frac{(2r-1)t_1(x)t_{r-1}(x) - (r-1)\left\{1\frac{(r-1)^2}{N^2}\right\}t_{r-2}(x)}{P} \tag{4}$$

When the above equation is normalized, we will get the squared form $\rho(r,N)$ which is written as follows:

$$\rho(r,N) = \frac{N\left(1 - \frac{1}{N^2}\right)\left(1 - \frac{2^2}{N^2}\right)\left(1\frac{r^2}{N^2}\right)}{2r + 1} \tag{5}$$

Chebyshev moment can be calculated in different order $r$ and repetition $s$ that accumulates 100 moment features that is extracted from each segmented image. Therefore, up to 900 features can be extracted from single input image considering the target, shadow and shadow + target images of three different segment methods.

## Residual network 152-CNN

Ever increasing complexity of images has made it vital to incorporate more complex models to extract the contents to be recognized from such images, thus architecture like CNN also suffers with such complexity images force the researcher to adapt such complex models like RENET50, RESNET102 and RESNET152 to overcome performance degradation problem of CNN due to sensitivity and over fitting issues and challenges.

Resnet152 has much lower complexity and delivers better accuracy it reduces 256 dimensions into 64 dimensions on 1*1 convolutions, then performs recovery through 1*1 convolutions again, i.e., the process of both dimension reduction and dimension increase carried out. Figure 5 represents the RESNET152 with three layers deep adapted from [37]. For our study, we selected RESNET152 to overcome the issues faced by plain networks such as AlexNet, ZFNet and VGGNet. When these plain networks gets deeper, it suffers with vanishing and exploding gradient issues whereas in case of RESNET skip/shortcut connection is added between input and output after few weight layers. Based on ILVRC study report, we used RESNET152 instead of RESNET50 or RESNET101 [38]. We wanted to have a network which is deeper (152) and anything beyond that is difficult to control in terms of time complexity since the researcher is a beginner not an expert.

## Long- and short-term memory

It is a cyclic network that controls all types of information flow in a network, also updates the existing memory of the information network. It is also efficient in rectifying the problem of gradient disappearance in Back-Propagation through Time (BPTT) training. LSTM model consist of one storage unit and 3 control gates.

**Input Gate.** The input to the LSTM consists of hidden state of the previous time $H_{t-1}$ and newly received data at the current time $X_t$. To get the final output of this input gate $I_t$, *tanh*

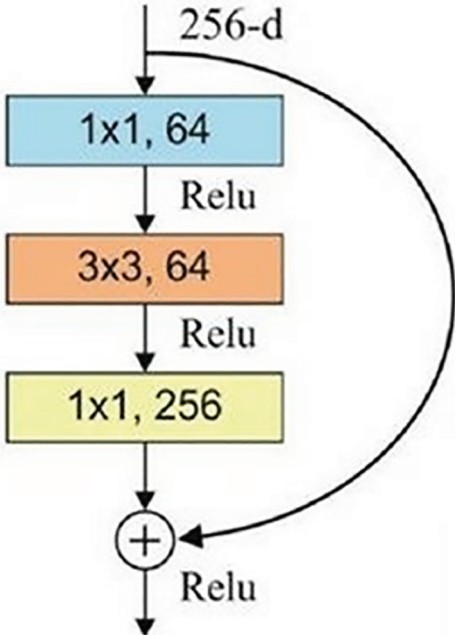

**Fig 5. RESNET152 3-layer residual elements [37].**

function combined with these two input variables, therefore, with the activation function $\sigma$ , $I_t$ is written as,

$$I_t = \sigma \left( X_t W_{xi} + H_{t-1} W_{hi} + b_i \right) \qquad (6)$$

where $b_i$ represents the new input value that is added to previous input, $W_{xi}$ denotes the weight of input and $W_{hi}$ denotes weight of input at hidden layer.

**Oblivion gate.** This gate is used to forget the current node's historical information, by doing so redundant and duplicate historical information from the network is removed. Moreover, capacity information and associated information will be redefined. When the resulting information of the function is closed, then the value of the information is higher than before, thus, it keeps it, unless the value will be discarded. Therefore, the function of forgetting features at time $t$, $F_t$ is written as,

$$F_t = \sigma \left( X_t W_{xf} + H_{t-1} W_{hf} + b_f \right) \qquad (7)$$

where $b_f$ represents new or current value is added when the old value that the network is forgetting gets replaced. $W_{xf}$ represents the weight of forgetting feature value and $W_{hf}$ represents the weight of forgetting feature value at hidden layer.

**Output gate.** The main objective of this gate is to have control over the output of nodes and if the node information includes main feature of an image, then the accuracy of the output will also improve. Therefore, it determines the output of previous memory update as to have control over the size of next output. The function is written as

$$O_t = \sigma \left( X_t W_{xo} + H_{t-1} W_{ho} + b_o \right) \qquad (8)$$

where $b_o$ represents the new output value is added when output of previous memory is lower. $W_{xo}$ represents the weight of current output feature value and $W_{ho}$ represents the weight of current output feature value at hidden layer.

**Memory cells.** This part of LSTM is used to store the state information, thus, maintaining long term historical information of various states has it come across. Candidate memory cells information needs to be given preference during calculation similar to above 3 gates, the calculation is performed but the difference is *tanh* function is in the range of [−1,1] is considered for activation function. The new candidate value $C_t$

$$\check{C}_t = \tanh \left( X_t W_{xc} + H_{t-1} W_{hc} + b_c \right) \tag{9}$$

In general, $b$ represents peepholes to let the gate layers to look at the cell state and these peepholes added to every gate. $W_{xc}$ represents the weight of new candidate current feature value and $W_{hc}$ represents the weight of new candidate current feature value at hidden layer.

The flow of information in hidden state is controlled through input gate, forget gate and output gate, and this flow is generally realized by using multiplication by elements with symbol $\odot$ [18]. Figure 6 represents the proposed Schematic Structures of LSTM Units. The information of upper time step memory cell and current time step candidate memory cell is combined to calculate current time step memory cell, i.e., $H_t \in R^{n*h}$ , which also controls the flow of information through forgetting door and input door. For this, the equation can be written as,

$$C_t = F_t \odot C_{t-1} \odot + I_t \odot \check{C}_t \tag{10}$$

## Combination of CNN and LSTM

Our proposed work mainly focuses on two important models: CNN with RESNET152 and LSTM to mainly deal with the feature extraction of dynamic facial expression of an image dataset, and to truly identify the recognition of that expression. This combination was proposed since CNN is very effective on extracting local information from images from the previous layer and LSTM is effective in dealing with time-series of input sequence, i.e., large or

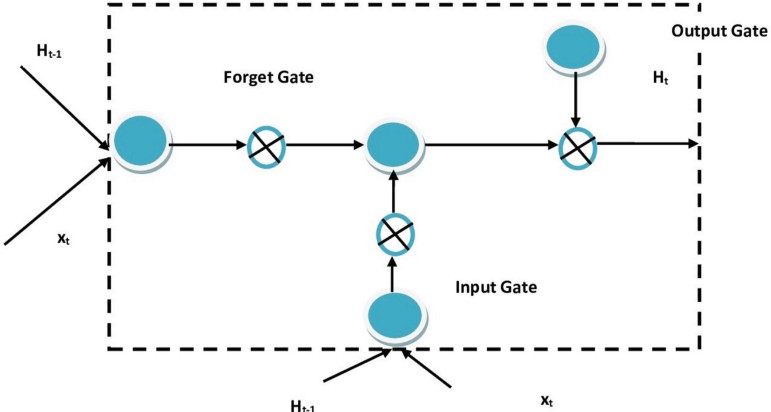

**Fig 6. Proposed schematic structures LSTM units.**

whole input sequence. The rich features obtained by CNN are fed to LSTM to determine or understand the dependencies that are relevant to the whole input image sequence. Figure 7 represents the Schematic Structures of the Proposed Work.

The steps of our proposed algorithm are given below:

**Algorithm 1:** Hybrid RESNET152-CNN-LSTM with Chebyshev Moment and K-fold Validation

**Step 1:** Obtain the input image from FER2013 dataset
**Step 2:** Extract them into $'N'$ frames
**Step 3:** Extract Features using Chebyshev Moment

$$K_{rs} = \frac{1}{2\pi\rho(\rho,m)} \sum_{p=0}^{m-1} \sum_{\theta=0}^{2\pi} t_\rho(p) * e^{-js\theta} * f(p,\theta)$$

$$\rho(r,N) = \frac{N\left(1-\frac{1}{N^2}\right)\left(1-\frac{2^2}{N^2}\right)\left(1\frac{r^2}{N^2}\right)}{2r+1}$$

**Step 4:** Extract the Corresponding $'n'$ CNN features
**Step 5:** Connect CNN with LSTM

$$I_t = \sigma\left(X_t W_{xi} + H_{t-1} W_{hi} + b_i\right)$$
$$F_t = \sigma\left(X_t W_{xf} + H_{t-1} W_{hf} + b_f\right)$$
$$O_t = \sigma\left(X_t W_{xo} + H_{t-1} W_{ho} + b_o\right)$$
$$\check{C}_t = \tanh\left(X_t W_{xc} + H_{t-1} W_{hc} + b_c\right)$$

**Step 6:** Form featuring layer of LSTM by connecting upper $'n'$ layer and lower $'N'$ layer of it

$$O_t = \sigma\left(X_t W_{xo} + H_{t-1} W_{ho} + b_o\right)$$

**Step 7:** Apply K-fold Validation
**Step 8:** Apply Multi-Library SVM for classification
Key part of the model is: The CNN model used in this study is RESNET152

Since, we are using RESNET152 model, the time complexity of our proposed algorithm must be kept simple. Basically, RESNET model uses bottleneck design to minimize the time complexity. It is achieved by adding 1*1 convolution at the start and end of the network (refer Figure fig5). For any CNN, the time complexity largely depends on number of filters, their

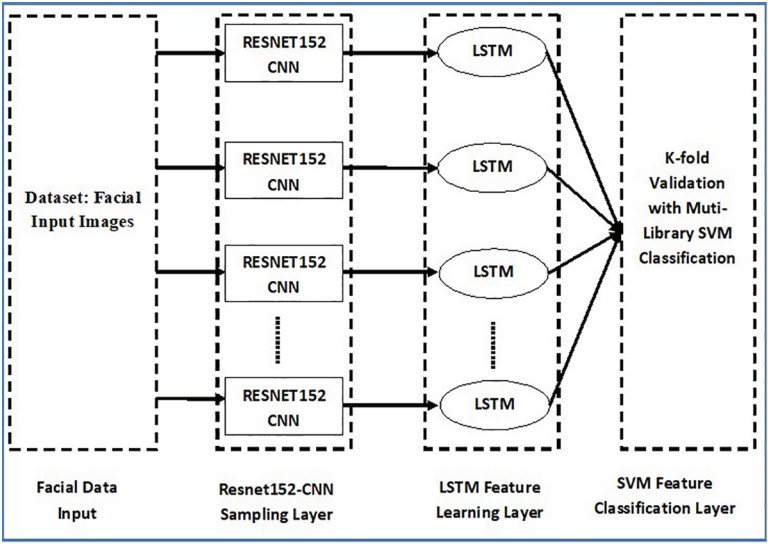

**Fig 7. Schematic structures of the proposed work.**

dimensions, and the dimensions of the input. However, the actual running time of a neural network can be significantly affected by some other factors such as the efficiency of the implementation and the hardware used for running the model. Overall, the time complexity of the network model not only involves forward propagation of data through the network, but also includes the back propagation of errors and repeated adjustments of the network's weights[38].

## K-fold class validation

K-nearest neighbors (KNN) is widely used for many network models to identify the nearest facial features that are present in local region with no class identification as it cannot identify global features, thus, validation is not in place. With such limitation of KNN, We have used k-fold class validation in which each individual subject is treated as separate class (5 classes of emotions in our work). Generally, this method is used to validate and evaluate the performance of the proposed or any model that is already been developed. The following steps are performed in this method:

(1) Divide the chosen dataset into 'k' equally sized subsets
(2) Then, the model is trained in k-1 subsets and evaluated on the remaining subsets
(3) Repeat the process for k times (till each subset is used once as evaluation subset)

Pseudo-code: For each subset $i$ from 1 to $k$,

(a) Train the model for all subsets of the dataset except the $i^{th}$ subset
(b) Evaluate the model on the $i^{th}$ subset and record the evaluation metric (accuracy and etc)
(c) Finally, calculate the average of the k evaluation metrics to get a final estimate of the model's performance

k-fold validation is considered to be a powerful technique since it allows the researcher to train and evaluate the model on all of the data under study, also retains few subsets for the purpose of evaluation. Thus, it helps in reduce over-fitting as the model's evaluation done with the data it has not come across before. More importantly, this method is not over sensitive to any particular class of a dataset and also ensures robustness of evaluation metrics.

Multi-library SVM: In our work, we have employed Multi-lib Support Vector Machine, a standard library for SVM for classification purposes. SVM is less prone to over-fitting problem as compared to other classifiers such as naïve base, but multi-class SVM contains separating hyper plane between the different classes of the target image is much easier and accurate in classifying the facial expressions. MATLAB Pattern Recognition Toolbox (PRTools) used for running all the codes and in all the experiments Radial Basis Kernel (RBF) is applied.

## Experimentation results

As mentioned in Dataset Preparation, FER 2013 dataset is manually reduced to 7074 images through manual selection, in which a total of five expression classes included. These 7074 images are used for test.The manual filtering is mainly performed to avoid images with occlusion, low quality and visual characteristics. Thus, the reduced dataset images will have only images with medium to high quality images that could lead to better detection of facial expressions as far as model's performance is concerned. As mentioned earlier in this paper, we used dataset from [13].

Table 3 and Figure 8 represents the class distribution of FER2013 dataset images that are manually filtered for this study. The drowsiness or closed eyes are not considered in this research work and with relevant datasets our future work might consider these expressions as well. The relationship between the facial expression and sleepiness should be an interesting scenario to study and it might impose some challenges as well.

## Simulation environment

MATLAB (Mat Laboratory) 2021a release version of the simulation software is used for the implementation and analysis of our proposed work, and due to the high speed training of Graphic Processing Unit (GPU), deep learning related tool boxes were initialized before-hand. The proposed method was compared with two other existing approaches, namely, CNN-SVM, ANN-LSTM in terms of accuracy, recall and precision.

## Evaluation metrics

Initially, our proposed approach is compared with three other models and the resulting performance is compared in terms of accuracy. MATLAB2021a is used for testing the performance of the model and with the accuracy performance value further analysis is done. The models considered for performance comparisons are AlexNet, VGG19 and ResNet50 and the accuracy values are given in Table 4.

To select the best model for this proposed work, we compared above mentioned 4 models for training and testing FER2013 dataset to determine the best model. By looking at the Figures 8–10 it is understood that RESNET152 model has outplayed other three models in terms of accuracy and Alexnet consumed minimum time to complete both training and testing the than other three models with undertaken FER2013 dataset. Figure 9 represents the accuracy of all four models while training the FER2013 dataset, Figure 10 represents the accuracy all four models while testing the trained FER2013 dataset and Figure 11 presents

**Table 3. Number of images considered from FER2013 dataset for this study**

| Image experimentation category | Happy | Angry | Fear | Neutral | Sad |
|---|---|---|---|---|---|
| Test | 2477 | 966 | 859 | 1466 | 1326 |

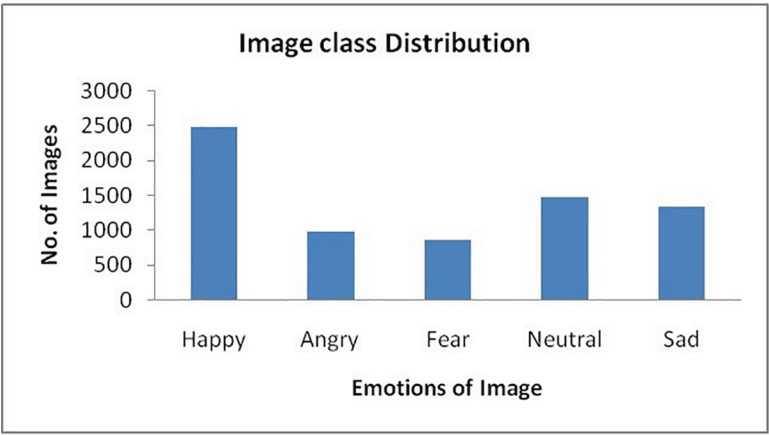

**Fig 8. Facial expression class distribution of FER2013 dataset.**

**Table 4. Accuracy of existing models and proposed in terms of classifier**

| Model Used | Accuracy while training (%) | Accuracy while testing (%) | Time taken (in seconds) |
|---|---|---|---|
| AlexNet | 74.22 | 75.12 | 1478 |
| VGG19 | 86.17 | 88.19 | 4736.5 |
| ResNEt50 | 85.62 | 87.75 | 2489 |
| RESNET152 | 89.92 | 90.23 | 2412 |

the time consumed by each model to complete execution are presented with respective values from Table 4. It is evident from Figure 8 that VGG19 has outperformed both AlexNet and ResNet50, but ResNet152 provided better accuracy while both training and testing. ResNet152 has achieved 89.92% and 90.23% of accuracy and time taken to achieve this is just 2412 seconds which is second minimum after alexnet than the other models considered.

**Accuracy** Accuracy is determined by using the following equation:

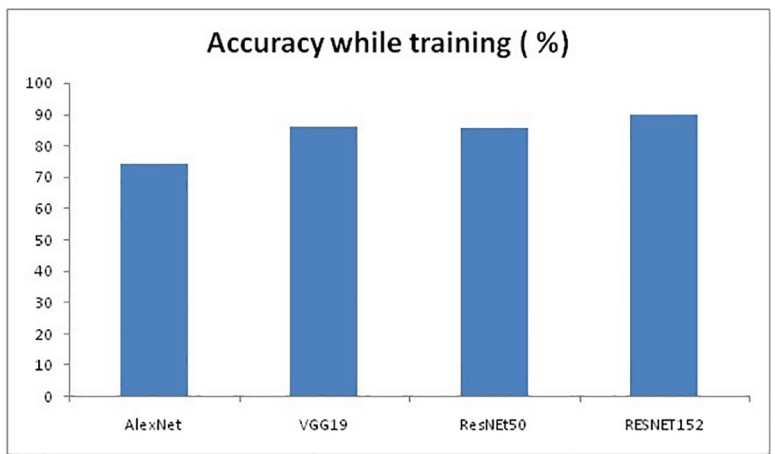

**Fig 9. Accuracy of models while training.**

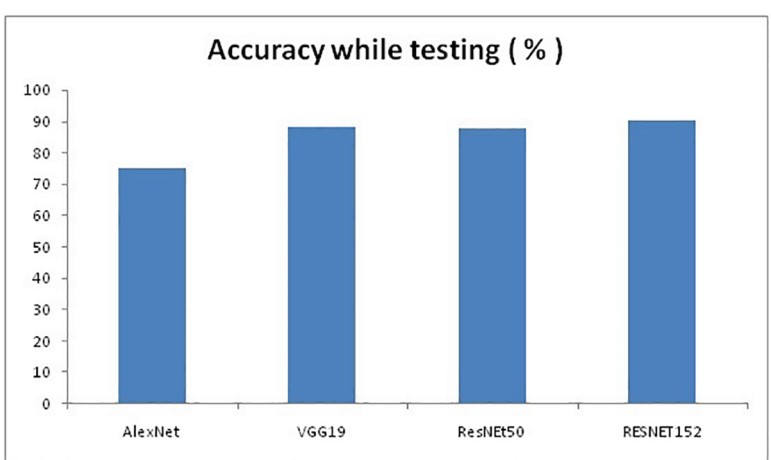

**Fig 10. Accuracy of models while testing.**

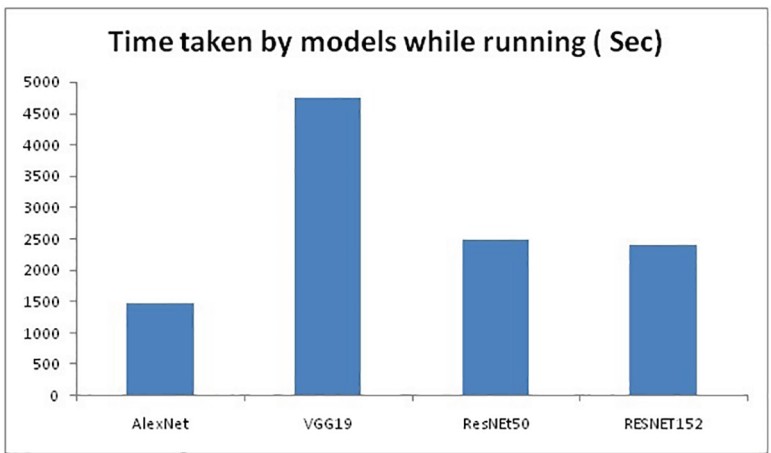

**Fig 11. Time taken by models while running.**

$$\text{Accuracy} \ = \frac{\text{Number of Correct Predictions}}{\text{Total number of Predictionn}}, \tag{11}$$

**Confusion matrix** It is a matrix that summarizes the performance of a classification model used in this work which aims to predict a categorical label for each input instance where N is the total number of target classes (5 emotions). Basically, this matrix compares the actual target values with those predicted by learning model used in this study.

True Positive (TP): correct prediction of an emotion
True Negative (TN): correct prediction of an incorrect emotion
False Negative (FN): incorrect prediction of an incorrect emotion
False Positive (FP): incorrect prediction of an emotion

Table 5 presents the basic structure of confusion matrix used in our research.
Figure 12 represents the confusion matrix for five emotions in which red represents happy, blue represents sad, angry, neutral or fear and yellow represents the prediction is not sad, angry, neutral or fear. Finally, the green represents that the image was not happy but predicted to be happy. Recall, precision is calculated with the help of confusion matrix using TP, TN, FP and FN. Recall is estimated using the following equation,

$$\text{Recall} \ = \frac{TP}{TP + FN} \tag{12}$$

**Table 5. Confusion matrix**

| Predicted class | Actual Class | | |
|---|---|---|---|
| | | Positive | Negative |
| | Positive | TP | FP |
| | Negative | FN | TN |

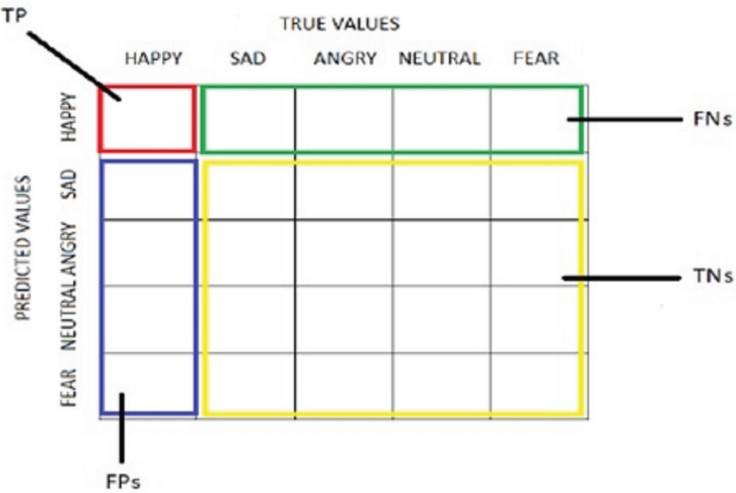

**Fig 12. Confusion matrix for five emotions [9].**

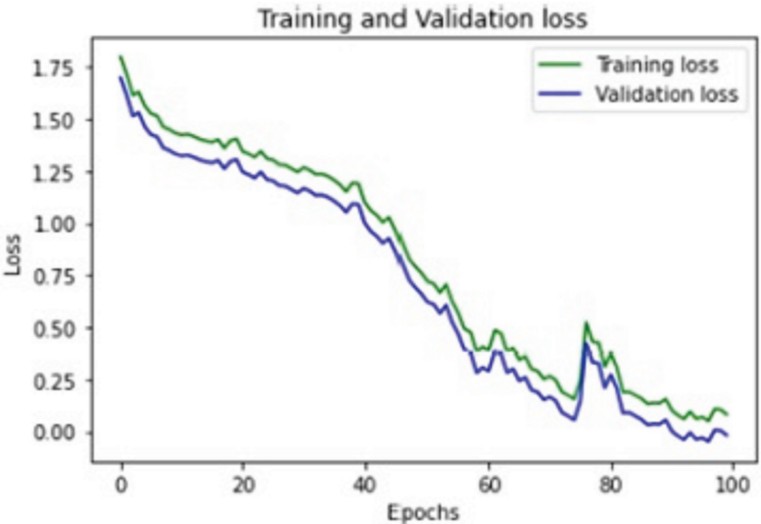

**Fig 13. Training and validation loss of proposed approach.**

Precision is calculated using the following equation,

$$\text{Precision} = \frac{TP}{TP + FP} \tag{13}$$

The training and validation (k-fold) loss functions of our proposed approach were also calculated to increase the accuracy performance with the help of training and evaluating results. The training and validation loss and accuracy of our proposed approach with RESNET 152 is represented in Figures 13 and 14. Table 6 represents the metric values of proposed approaches and two other existing methods CNN-SVM, ANN-LSTM. It is evident from the table that the

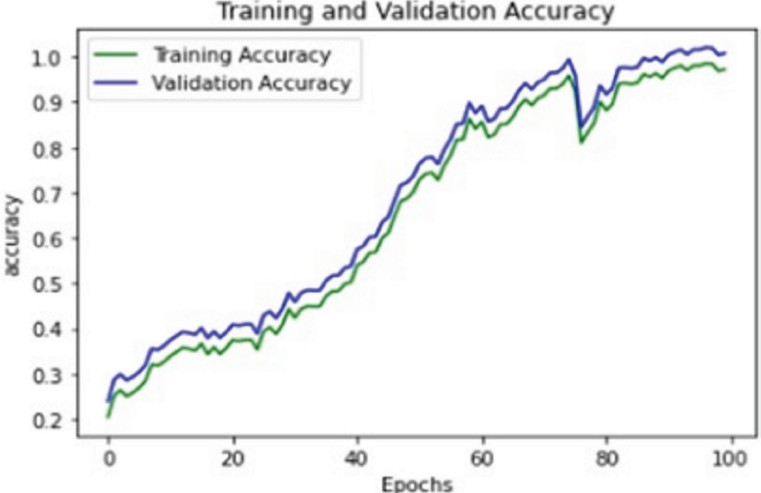

**Fig 14.  Training and Validation accuracy of proposed approach.**

**Table 6.  Matric values of CNN-SVM, ANN-LSTM and proposed approaches**

| Metrics/Methods | Accuracy | Recall | Precision | Sensitivity | F1 Score |
|---|---|---|---|---|---|
| CNN-SVM | 63% | 63% | 87% | 72% | 76% |
| ANN-LSTM | 69% | 76% | 86% | 79% | 82% |
| Proposed Approach | 82% | 84% | 92% | 87% | 91% |

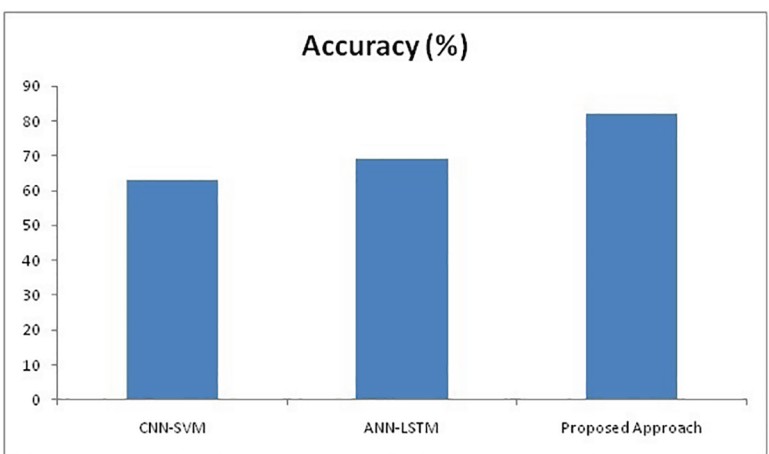

**Fig 15. Comparison of accuracy metric with different methods.**

proposed approach has outplayed other two approaches in terms of accuracy, recall, precision, Sensitivity and F1 Score metrics.

From Figure 15, it is evident that the percentage of accuracy achieved by CNN-SVM stands on 63%, while ANN-LSTM stands at 69%, but our proposed approach has achieved 82% that is 19% higher than CNN-SVM and 13% higher than ANN-LSTM

For the recall (Figure 16), the proposed method has provided much more improved numbers than the other two approaches with 84% where CNN-SVM and ANN-LSTM stood at

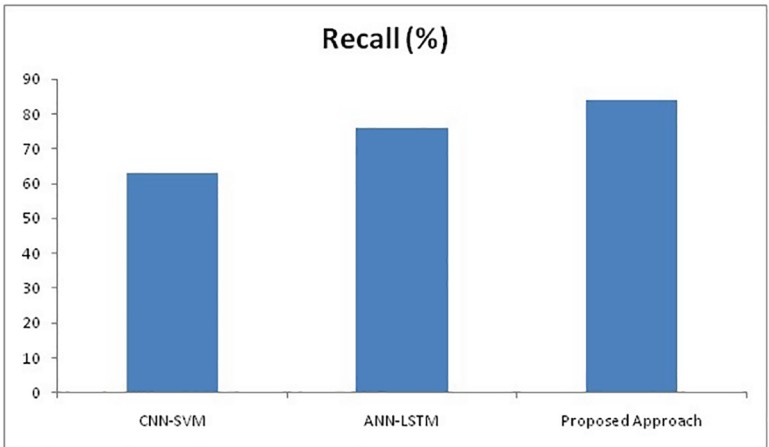

**Fig 16. Comparison of recall metric with different methods.**

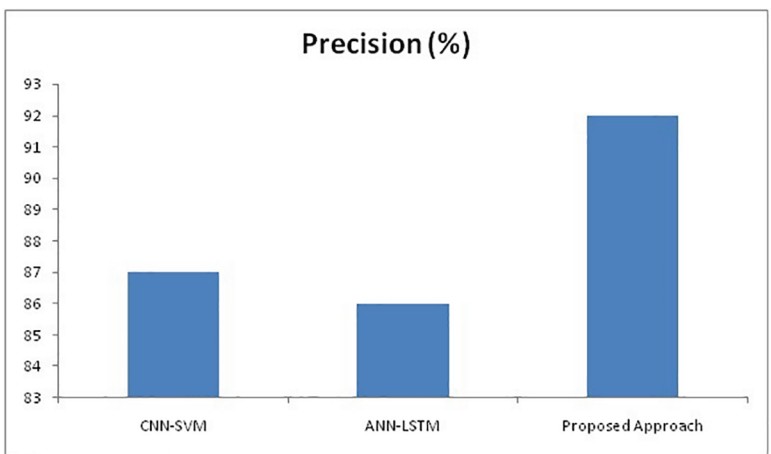

**Fig 17. Comparison of precision metric with different methods.**

63% and 76% respectively. For precision (Figure 17), CNN-SVM and ANN-LSTM produced 87% and 86% respectively while the proposed approach able to produce 92

Figure 18 depicts the sensitivity percentage of all methods which proves that the proposed approach has produced better results with 87% whereas CNN-SVM and ANN-LSTM achieved 72% and 79% respectively. Figure 19 depicts the F1 score of all approaches where proposed approach has achieved 91% which is higher than other two methods with 76% and 82%. Therefore, the proposed approach has delivered far better results than the other two approaches in all aspects and delivered better performance in detecting facial expressions recognition of the undertaken FER2013 dataset.

## Discussion and findings

Having many researches undertaken on CNN-LSTM based mechanism for facial expression recognition; our proposed approach does well in identifying around 100 features using Chebyshev Moment from a single image in which 62 are treated as important features that

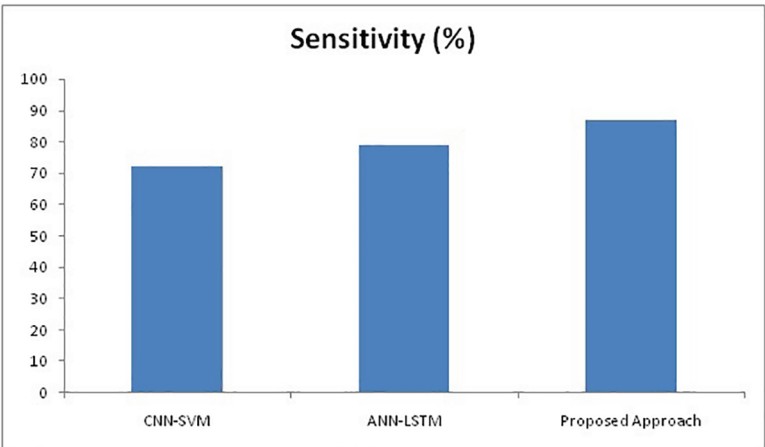

**Fig 18. Comparison of sensitivity metric with different methods.**

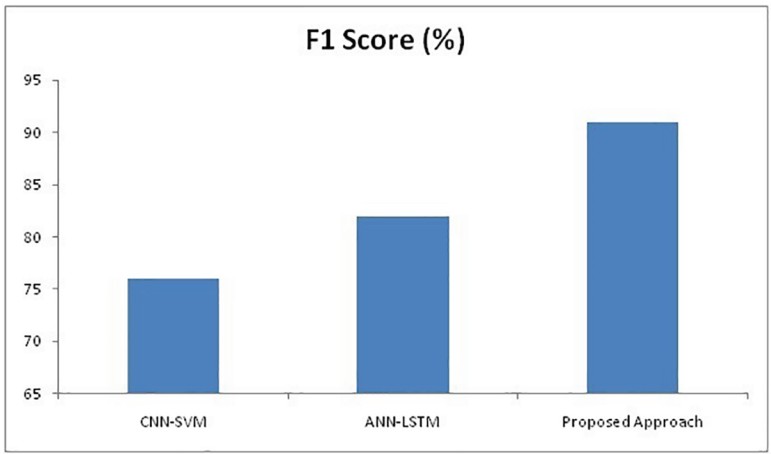

**Fig 19. Comparison of F1 Score metric with different methods.**

are again categorized as primary and secondary features (Figure 1). This helps the model to accumulate as many features as possible from the input image. In addition, we also employed Resnet252 CNN for better efficacy results along with K-fold validation which can classify the image based on the classes defined for expressions present on the image. Once validation is performed, Multi-library SVM will take care of the classification based on the K-fold validation output. More importantly, our study has only considered 7074 images from FER2013 dataset among 35000 images based on manual filtering. Hence, the proposed hybrid model has delivered better results than the traditional benchmarking approaches that are considered for the comparison in this study.

## Conclusion and future enhancement

In this paper, we have proposed RESNET152-CNN-LSTM based hybrid model to detect facial expressions or emotions of human images, for which we have used manually filtered 7074 images from FER2013 dataset. The proposed hybrid model has used CNN-REsNet152 model

that is incorporated with Chebyshev Moment and K-fold validation techniques for improving feature extraction, and evaluating the validation loss and validation accuracy respectively. The proposed approach has been compared with CNN-SVM and ANN-LSTM methods in terms of accuracy is at 82%, Recall 84%, Precision 92%, Sensitivity 87% and F1 Score 91% metrics. The proposed approach has produced much improved results than other two approaches. For further research, Attention Mechanism (AM) can be considered to analyze occluded and low quality images with CNN-LSTM hybrid model, and some other additional techniques and additional datasets such as CK+ will be added along with FER2013.

Expressions are not always to be explicit, especially with half or fully closed eyes denote that the driver is either drowsy or extremely restless, i.e., especially cars and aircrafts. For further research, Attention Mechanism (AM) can be considered to analyze occluded and low quality images with CNN-LSTM hybrid model, and some other additional techniques and datasets such as CK+, RAF-DB and FER Plus along with FER2013 can be added. The most commonly used classifiers like softmax will be considered for our future work. This combination was proposed since CNN is very effective on extracting input image's local information from its previous layers and LSTM is effective in dealing with time-series of input sequence, i.e., large or whole input sequence. The rich features obtained by CNN are fed to LSTM to determine or understand the dependencies that are relevant to the whole input image sequence.Transformer based techniques such as Vision Transformer (ViT) and Self Attention Based SWIN transformer could be considered in our future research when dealing with cameras and videos to ensure safety of peoples who use road surfaces.

## Author contributions

**Conceptualization:** Samanthisvaran Jayaraman.

**Supervision:** Anand Mahendran.

**Writing – original draft:** Samanthisvaran Jayaraman.

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
