## [Decision Letter · Decision Letter 0]

11 Nov 2024

PONE-D-24-19578CNN-LSTM based Emotion Recognition using Chebyshev Moment and K-fold Validation with Multi-Library SVMPLOS ONE

Dear Dr. Mahendran,

Thank you for submitting your manuscript to PLOS ONE. After careful consideration, we feel that it has merit but does not fully meet PLOS ONE’s publication criteria as it currently stands. Therefore, we invite you to submit a revised version of the manuscript that addresses the points raised during the review process.

**ACADEMIC EDITOR: **

**Thank you for submitting your manuscript to PLOSE ONE. We have received feedback from reviewers on your paper. While the reviewers acknowledge the potential of your work, they have raised several concerns that need to be addressed. In light of these comments, we invite you to submit a revised version of your manuscript that addresses the reviewers' concerns for further consideration. Please include a detailed response to each point raised by the reviewers, and clearly indicate the changes made in the manuscript.**

We look forward to receiving your revised manuscript.

Kind regards,

Qionghao Huang

Academic Editor

PLOS ONE

**Journal Requirements:**

3. Please note that your Data Availability Statement is currently missing the repository name. If your manuscript is accepted for publication, you will be asked to provide these details on a very short timeline. We therefore suggest that you provide this information now, though we will not hold up the peer review process if you are unable.

5. We note that Figure 7 includes an image of a participant in the study. 

Reviewers' comments:

Reviewer's Responses to Questions

**Comments to the Author**

1. Is the manuscript technically sound, and do the data support the conclusions?

Reviewer #1: Partly

Reviewer #2: Yes

2. Has the statistical analysis been performed appropriately and rigorously? 

Reviewer #1: Yes

Reviewer #2: Yes

3. Have the authors made all data underlying the findings in their manuscript fully available?

Reviewer #1: Yes

Reviewer #2: Yes

4. Is the manuscript presented in an intelligible fashion and written in standard English?

Reviewer #1: Yes

Reviewer #2: Yes

5. Review Comments to the Author

**Reviewer #1:** The study proposes a hybrid CNN-LSTM model using RESNET152 CNN and Multi-Library SVM for facial emotion recognition. The model incorporates Chebyshev moments for feature extraction and K-fold validation for performance evaluation. The approach aims to improve accuracy, recall, and precision compared to existing methods. Some issues need to be addressed in the following:

- Research Motivation: The motivation behind the study is not clearly articulated. The paper should explicitly state the research gap that the proposed method aims to address. It is recommended that the authors clearly and concisely present this aspect.

- Recent Works: In the related work section, it would be beneficial to include some recent related works such as ‘Face2nodes: learning facial expression representations with relation-aware dynamic graph convolution networks, INS, 2023’, ‘FER-CHC: Facial expression recognition with cross-hierarchy contrast, ASOC, 2023’, ‘Emotion recognition from large-scale video clips with cross-attention and hybrid feature weighting neural networks, ERPH, 2023’, and ‘Facial expression recognition with grid-wise attention and visual transformer, INS,2021’. This will provide readers with a more current understanding of the field.

- Choice of Network Modules: Most current models use Vision Transformers (ViT) as the backbone for visual tasks. The authors should explain why they chose CNN or LSTM for designing the network. What are the advantages of these network modules compared to the existing mainstream modules? It is suggested that the authors supplement this part with relevant explanations.

- Chebyshev Moments: The motivation for introducing Chebyshev Moments is unclear. Are there better alternatives in the same category? Additionally, the experimental section does not provide a good comparison with similar methods. It is recommended to justify this part.

- Datasets and Generalization: In the experimental section, it is suggested to include more common datasets to demonstrate the generalization of the proposed method on different models. Additionally, compare the proposed model with the latest models to highlight its advancements.

- Ablation Studies: It is recommended to conduct comprehensive ablation studies to show the impact of different modules on the model’s performance. Additionally, perform parameter sensitivity analysis to demonstrate the effect of different parameters on the model’s performance.

- Image Quality: The quality of the images in the paper is relatively low. It is suggested to optimize them, preferably using vector graphics.

**Reviewer #2: **1- Author has mistakenly written vgg9 in the graph and tables instead vgg19. Kindly correct it.

2-Could you elaborate on the specific criteria used for the manual filtering of the FER2013 dataset? How were factors such as contrast, brightness, and image angle quantitatively assessed during this process?

3-Why did you choose the Gaussian Filter and Chebyshev Moment for feature extraction? How do they compare to other commonly used feature extraction methods in the context of facial expression recognition?

4-In your training methodology, you combined CNN, LSTM, and ResNet152. Could you provide more insights into how these architectures complement each other in this specific application? What advantages does the combined model offer compared to using CNN or ResNet152 alone?

5-What is the advantage of using a Multi-library SVM classifier as the final classification step? How does it perform compared to traditional SVM or other classification methods like softmax?

6-Why did you manually select and filter the dataset? On what specific criteria were the images discarded, and what were the reasons for eliminating certain images?

7-Please improve the quality and symmetry of figures no 9, 12, 13, and 14.

8- The purpose of your paper is not clearly stated in the abstract. Please revise the abstract to indicate that the paper focuses on analyzing the driver's facial expressions to determine their mood or emotional state while driving. This will help readers quickly understand the core objective of your study and its relevance to driver behavior monitoring and safety.

9- What is the connection between facial expressions and the driver's sleepiness? Are closed eyes considered part of facial expressions in your analysis, or are they treated separately? How do you address the relationship between facial expressions and drowsiness in your study?

10- Many references in the paper are outdated. Please update the references to include more recent sources, preferably from 2020 to 2024, to ensure the research is aligned with the latest developments in the field

6. PLOS authors have the option to publish the peer review history of their article (what does this mean?). If published, this will include your full peer review and any attached files.

Reviewer #1: No

Reviewer #2: No

---

## [Author Response · Author response to Decision Letter 1]

23 Dec 2024

All the Reviewers comments have been made and separate document regarding the Response to Reviewers file

---

## [Decision Letter · Decision Letter 1]

14 Jan 2025

PONE-D-24-19578R1CNN-LSTM based Emotion Recognition using Chebyshev Moment and K-fold Validation with Multi-Library SVMPLOS ONE

Dear Dr. Mahendran,

Thank you for submitting your manuscript to PLOS ONE. After careful consideration, we feel that it has merit but does not fully meet PLOS ONE’s publication criteria as it currently stands. Therefore, we invite you to submit a revised version of the manuscript that addresses the points raised during the review process.

**ACADEMIC EDITOR: **Thank you for submitting your manuscript to PLOS ONE. We have received feedback from reviewers on your paper. The reviewers acknowledge the improvements made in your work and appreciate the potential it holds. However, they have identified a few areas that require further refinement. In light of these comments, we invite you to submit a revised version of your manuscript that addresses the reviewers' concerns for further consideration. Please include a detailed response to each point raised by the reviewers, and clearly indicate the changes made in the manuscript.

We look forward to receiving your revised manuscript.

Kind regards,

Qionghao Huang

Academic Editor

PLOS ONE

Journal Requirements:

Reviewers' comments:

Reviewer's Responses to Questions

**Comments to the Author**

1. If the authors have adequately addressed your comments raised in a previous round of review and you feel that this manuscript is now acceptable for publication, you may indicate that here to bypass the “Comments to the Author” section, enter your conflict of interest statement in the “Confidential to Editor” section, and submit your "Accept" recommendation.

Reviewer #1: All comments have been addressed

Reviewer #3: (No Response)

2. Is the manuscript technically sound, and do the data support the conclusions?

Reviewer #1: (No Response)

Reviewer #3: (No Response)

3. Has the statistical analysis been performed appropriately and rigorously? 

Reviewer #1: (No Response)

Reviewer #3: (No Response)

4. Have the authors made all data underlying the findings in their manuscript fully available?

Reviewer #1: (No Response)

Reviewer #3: (No Response)

5. Is the manuscript presented in an intelligible fashion and written in standard English?

Reviewer #1: (No Response)

Reviewer #3: (No Response)

6. Review Comments to the Author

Reviewer #1: The author's response has addressed my concerns on this paper, and I recommend an acceptance of this paper.

Reviewer #3: 1. The Introduction contains excessive content focused on automotive driving.

2. Line 39 and 70 lack proper punctuation at the end.

7. PLOS authors have the option to publish the peer review history of their article (what does this mean?). If published, this will include your full peer review and any attached files.

Reviewer #1: No

Reviewer #3: No

---

## [Decision Letter · Decision Letter 2]

13 Feb 2025

CNN-LSTM based Emotion Recognition using Chebyshev Moment and K-fold Validation with Multi-Library SVM

PONE-D-24-19578R2

Dear Dr. Mahendran,

We’re pleased to inform you that your manuscript has been judged scientifically suitable for publication and will be formally accepted for publication once it meets all outstanding technical requirements.

Kind regards,

Alessandro Bruno, Ph.D.

Academic Editor

PLOS ONE

Additional Editor Comments (optional):

Dear Authors,

You've worked out all the residual minor issues that were raised in the previous peer-review step.

Your paper is now ready for acceptance.

Kind regards,

A.B.

Reviewers' comments:

Reviewer's Responses to Questions

**Comments to the Author**

1. If the authors have adequately addressed your comments raised in a previous round of review and you feel that this manuscript is now acceptable for publication, you may indicate that here to bypass the “Comments to the Author” section, enter your conflict of interest statement in the “Confidential to Editor” section, and submit your "Accept" recommendation.

Reviewer #1: All comments have been addressed

Reviewer #3: (No Response)

2. Is the manuscript technically sound, and do the data support the conclusions?

Reviewer #1: (No Response)

Reviewer #3: (No Response)

3. Has the statistical analysis been performed appropriately and rigorously? 

Reviewer #1: (No Response)

Reviewer #3: (No Response)

4. Have the authors made all data underlying the findings in their manuscript fully available?

Reviewer #1: (No Response)

Reviewer #3: (No Response)

5. Is the manuscript presented in an intelligible fashion and written in standard English?

Reviewer #1: (No Response)

Reviewer #3: (No Response)

6. Review Comments to the Author

Reviewer #1: The author's response has addressed my concerns on this paper, and I recommend an acceptance of this paper.

Reviewer #3: (No Response)

7. PLOS authors have the option to publish the peer review history of their article (what does this mean?). If published, this will include your full peer review and any attached files.

Reviewer #1: No

Reviewer #3: No

---

## [Editor Report · Acceptance letter]

PONE-D-24-19578R2

PLOS ONE

Dear Dr. Mahendran,

I'm pleased to inform you that your manuscript has been deemed suitable for publication in PLOS ONE. Congratulations! Your manuscript is now being handed over to our production team.

Kind regards,

on behalf of

Associate Professor Alessandro Bruno

Academic Editor

PLOS ONE